# A novel role for the root cap in phosphate uptake and homeostasis

**Satomi Kanno**[1,2,3,4†], **Jean-François Arrighi**[2,3,4†‡], **Serge Chiarenza**[2,3,4], **Vincent Bayle**[2,3,4§], **Richard Berthomé**[5], **Benjamin Péret**[2,3,4¶], **Hélène Javot**[2,3,4], **Etienne Delannoy**[2,3,4**], **Elena Marin**[2,3,4], **Tomoko M Nakanishi**[1], **Marie-Christine Thibaud**[2,3,4*], **Laurent Nussaume**[2,3,4*]

[1]Graduate School of Agricultural and Life Sciences, University of Tokyo, Tokyo, Japan; [2]Laboratoire de Biologie du Developpement des Plantes, Institut de Biosciences et Biotechnology Aix-Marseille, Commissariat à l'Energie atomique et aux énergies alternatives, Saint Paul Les Durance, France; [3]UMR 7265 Biol. Veget. & Microbiol. Environ., Centre National de Recherche Scientifique, Saint Paul Les Durance, France; [4]UMR 7265, Aix-Marseille Université, Marseille, France; [5]Laboratoire des Interactions Plantes Micro-organismes, Castanet-Tolosan, France

**\*For correspondence:** mcthibaud@cea.fr (MCT); lnussaume@cea.fr (LN)

[†]These authors contributed equally to this work

**Present address:** [‡]Laboratoire des Symbioses Tropicales et Méditerranéennes, Montpellier, France; [§]Laboratoire de Reproduction et Développement des Plantes, Ecole Normale Supérieure de Lyon, Université Claude Bernard Lyon 1, Lyon, France; [¶]Biochimie et Physiologie Moléculaire des Plantes, Institut Claude Grignon, UMR 5004 CNRS/INRA/Supagro-M/UM2, Montpellier, France; [**]Unité de Recherche en Génomique Végétale, Université d'Evry Val d'Essonne, Evry, France

**Competing interests:** The authors declare that no competing interests exist.

**Abstract** The root cap has a fundamental role in sensing environmental cues as well as regulating root growth *via* altered meristem activity. Despite this well-established role in the control of developmental processes in roots, the root cap's function in nutrition remains obscure. Here, we uncover its role in phosphate nutrition by targeted cellular inactivation or phosphate transport complementation in *Arabidopsis*, using a transactivation strategy with an innovative high-resolution real-time [33]P imaging technique. Remarkably, the diminutive size of the root cap cells at the root-to-soil exchange surface accounts for a significant amount of the total seedling phosphate uptake (approximately 20%). This level of Pi absorption is sufficient for shoot biomass production (up to a 180% gain in soil), as well as repression of Pi starvation-induced genes. These results extend our understanding of this important tissue from its previously described roles in environmental perception to novel functions in mineral nutrition and homeostasis control.

## Introduction

Mineral nutrition has an essential function in plant roots and its spatial localization has been a subject of controversy for a long time. Following the invention of the compound microscope, pioneering descriptions of plant tissues in the seventeenth century by Marcello Malpighi (*Malpighi, 1675*) and Nehemiah Grew (*Grew, 1682*) established the bases of plant anatomy. Although both authors drew a connection between plant nutrition and distinct anatomical location, Malpighi proposed the root hairs whereas Grew proposed the root cap, at the distal end of the root tip. Our understanding of the spatial localization of plant nutrition has undergone great advances in the past twenty-five years. Combined efforts in molecular biology, genetics, biochemistry and electrophysiology have identified several key aspects to ion uptake, with a central role for plasma membrane (PM) ion transporters in this process. These transporters often belong to broad multigenic families (a likely consequence of their vital role) with overlapping expression patterns (*Nussaume et al., 2011*). However, this high redundancy has held back their genetic study to some degree, preventing analysis of ion uptake contribution at specific root localities.

The uptake of phosphate (Pi), an essential plant macronutrient, relies on a family of nine high-affinity transporters, identified as PHT1 in the plant model *Arabidopsis* (*Nussaume et al., 2011*). Mineral nutrition is classically associated with the root epidermis, and the various PHT1 members

**eLife digest** All plants need phosphate to grow because it is a major component of DNA and many other biological molecules. Most of the Earth's soil is poor in phosphate, and so farmland is routinely supplemented with fertilizers to provide crops with this essential nutrient. However, phosphate fertilizers are becoming scarce and their quality is expected to decline in the near future. Plant biologists must therefore determine how to adapt plants to a restricted supply of this resource, in order to sustain high crop yields for the growing world population.

Plants are known to absorb phosphate through specific protein-based transporters located in the cells that make up the outer layer of roots. These proteins are highly concentrated at the root tip, and while this specialized tissue is well-known for perceiving gravity and light, it had not been shown to play a role in phosphate absorption.

Kanno, Arrighi et al. have now used genetically modified Arabidopsis plants to demonstrate that phosphate can be taken up via the small cells that surround the root tip. The experiments showed that the absorbed phosphate rapidly reaches the leaves within minutes, helps the plant grow and modifies its metabolism. As the root tip can accumulate high amounts of phosphate in order to sustain its own activity, it was important to distinguish uptake of phosphate from the environment from redistribution of phosphate already within the plant. Kanno, Arrighi et al. tackled this issue through the development of a new radioactive micro-imaging technique.

Phosphate transporters are also present within the cell layers within the root, but their purpose and activity are not well described. Further studies are needed to analyze the role of other root cell layers in phosphate uptake and transport, and the newly developed techniques will help decipher the mechanisms involved.

were identified within this cell layer (*Mudge et al., 2002*; *Karthikeyan et al., 2002*; *Misson et al., 2004*; *Nussaume et al., 2011*). These transporters are particularly abundant where hair cells develop (*Daram et al., 1998*; *Brady et al., 2007*), as this increases the surface area in contact with the environment and improves nutrient uptake (*Peret et al., 2011*). Transcriptomic analyses of specific root cell layers (*Birnbaum et al., 2003*) and analysis of the PHT1 expression pattern has revealed the accumulation of these transporters in the root tip (*Mudge et al., 2002*; *Karthikeyan et al., 2002*; *Misson et al., 2004*). Nevertheless, determining how PHT1 proteins contribute to Pi uptake at the root tip is difficult, particularly since this area contains the primary root meristem. This tissue, which is essential for root growth, generates new cells that require Pi and its accumulation to build essential components including nucleic acids, ATP, and phospholipids. Consequently, this complicates differentiating Pi uptake from the Pi translocation (from epidermis) necessary to sustain active cell division in the root tip. Our approach to overcome this problem employs a recently developed high-resolution live radioisotope micro-imaging system (*Kanno et al., 2012*) combined with targeted cell ablation or complementation by genetic manipulation. These findings establish the importance of the root cap in phosphate uptake and homeostasis.

## Results

### Functional elements of Pi uptake are localized within the root cap

PHT1;1 and PHT1;4, the two most important high-affinity phosphate transporters required for up to 75% of Pi uptake in *Arabidopsis*, are localized to the distal part of the root tip (*Shin et al., 2004*). Both transporters are observed in the root tip (primarily in the root cap), where they are expressed at the transcriptional and protein levels (*Figure 1A*) with PHF1, a crucial component facilitating PHT1 post-translational regulation and its targeting to the PM (*Gonzalez et al., 2005*; *Bayle et al., 2011*) (*Figure 1A*). The root tip surrounds and protects the meristem, while also acting as the initial contact point between the roots and soil. In order to visualize Pi in plants, pulses of radioactive tracer were applied by immersing the whole root system in $^{33}$Pi solution. When fed to plantlets, $^{33}$Pi accumulation was observed at the extremity of the root tip (*Figure 1B*), including the meristematic zone (*Figure 1C*).

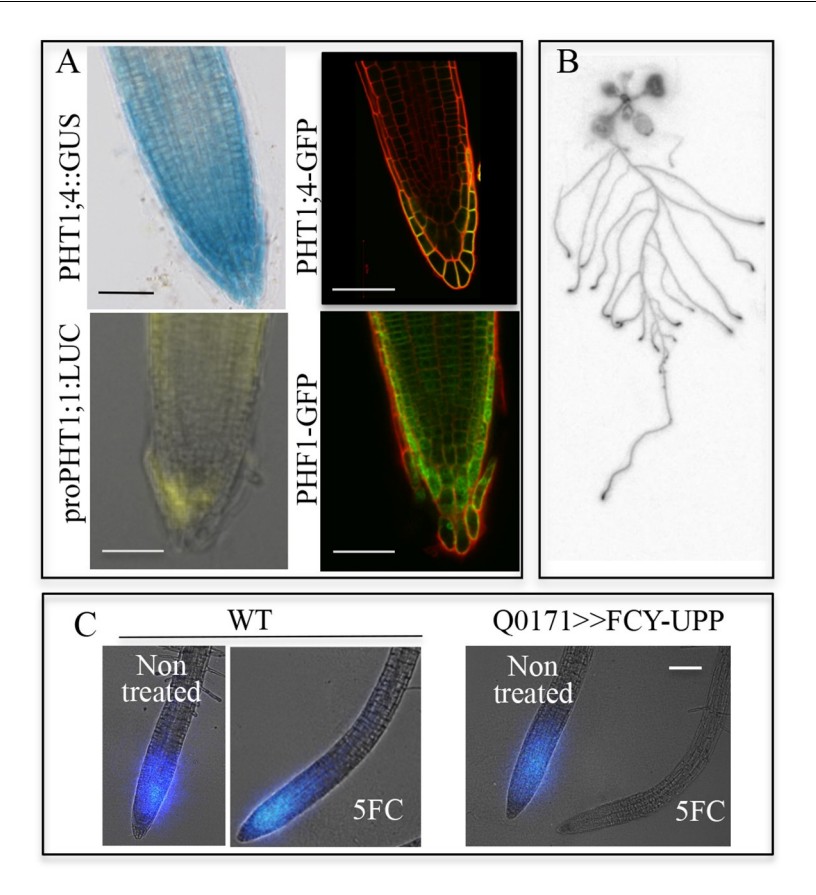

**Figure 1.** Active Pi transporters are localized in the root cap. (**A**) Reporter lines expressing transcriptional and translational fusions for the high affinity transporters expressed in the root (PHT1;1 and PHT1;4) are localized in the root cap, in addition to PHF1, a major post-translational regulator required for PHT1 targeting to the plasma membrane. Scale bars: 50 μm. (**B**) Accumulation of $^{33}$P in the root tip of Arabidopsis plantlets. Whole roots were immersed in $^{33}$P-enriched solution for 1 day. (**C**) Pi accumulation in the root tip is abolished by targeted ablation with 5FC in the Q0171>>FCY-UPP line. The short pulse of $^{33}$Pi applied to the WT and Q0171>>FCY-UPP lines was revealed by a live radioisotope microimaging system. Light transmission and $^{33}$P images are merged. $^{33}$P content is represented as color intensity. Scale bar: 100 μm.

The following figure supplements are available for figure 1:

**Figure supplement 1.** Assay for Pi translocation to the root tip.

**Figure supplement 2.** Conditional negative marker expression in the root cap.

**Figure supplement 3.** Effect of 5FC (3.8 mM) on primary root growth in the Q0171>>FCY-UPP line.

Pi translocation absorbed by the whole root system contributes to the presence of Pi in the root tip, as revealed by the accumulation of radiotracer in the root tip within 30 min of $^{33}$Pi application to the middle of the root system (*Figure 1—figure supplement 1*). Consequently, we chose genetic approaches to decipher the role of the root tip in Pi uptake, since the small size of this tissue prevented application of $^{33}$Pi at the root tip in a precise or exclusive manner.

To bypass the functional redundancy of Pi transporters (comprising 9 distinct loci in Arabidopsis) and their overlapping expression at the tissue and cell levels (*Nussaume et al., 2011*), we produced lines in which elimination of the root cap was activated. Previously, this was achieved by introducing a diphtheria toxin A-chain gene driven by a root cap promoter (*Tsugeki and Fedoroff, 1999*). However, due to the essential functions of the root cap in several physiological processes, these plants

exhibited roots with abnormal differentiation and very poor growth rate. Therefore, we employed a conditional approach using the negative marker *FCY* (cytosine deaminase; *Stougaard, 1993*) coupled with uracil phosphoribosyltransferase (*UPP*; *Tiraby et al., 1998*). These enzymes convert the innocuous 5-fluorocytosine (5FC) to cytotoxic 5-fluorouracil and 5-fluoroUMP (*Figure 1—figure supplement 2A*), which stop cellular activities. Genes encoding a *FCY-UPP* fusion were specifically expressed in root cap cells using the Arabidopsis GAL4/UAS binary expression system (*Laplaze et al., 2005*). The *FCY-UPP* construct under control of a UAS promoter was introduced in a GAL4 line (Q0171) selected for its specific root cap expression. This line also expresses GFP under the control of a UAS promoter (*Figure 1—figure supplement 2*), for visualizing tissues in which the transactivation took place (*Figure 1—figure supplement 2*). As expected, a five-day treatment with 5FC drastically reduced the expression of GFP in the root cap (*Figure 1—figure supplement 2*); the low GFP signal may result from high GFP protein stability or low residual expression. Root tip cell viability was not affected, as revealed by propidium iodide staining (PI; *Figure 1—figure supplement 2*). In living cells, PI staining is restricted to the cell wall (as in the non-control plants), whereas it is localized to nuclei in dead tissues, as in the heat-shocked WT control (see inserts in *Figure 1—figure supplement 2*).

Pi absorption was visualized using a real-time radioisotope imaging system developed for plant nutrient uptake studies (*Kanno et al., 2012*). Roots were treated with short pulses of $^{33}$Pi radiotracer and examined by live radioisotope microimaging. The ability of the root cap to promote Pi uptake is illustrated in *Figure 1C*. Expression of *FCY-UPP* in the root cap combined with 5FC treatment clearly abolished the $^{33}$Pi accumulation observed in either the WT or the transgenic line not treated with 5FC. Treating the WT line with 5FC had no effect on either plant growth (results not shown) or Pi accumulation, demonstrating its innocuous nature when *FCY-UPP* is not expressed (*Figure 1C*). This result establishes the existence of an unexpected active Pi uptake process at the root cap level.

Nevertheless, this *FCY-UPP*-based system cannot be used to quantify the physiological effect of root cap ablation on Pi status, since arrested root growth was observed within 2 days of 5FC treatment. This alters the root architecture and prevents any precise quantitative comparison with the WT control (*Figure 1—figure supplement 3*).

## Quantification of Pi uptake by the root cap

As stated above, the high redundancy of Pi transporters poses technical difficulties to investigating their roles in a selected tissue. We used a PHF1 mutant (*phf1-1*) to circumvent this obstacle, as this mutation strongly reduces PHT1 accumulation in the PM, resulting in a 70–80% reduction in Pi uptake (*Bayle et al., 2011*; *Gonzalez et al., 2005*). Consequently, the *phf1-1* mutant exhibits phosphate starvation traits in Pi-rich medium, although its growth is only slightly reduced. This offers an appropriate genetic tool for targeted PHF1 complementation to restore Pi uptake in specific tissues. Using the same GAL4/UAS system described above, we back-crossed the GAL4 enhancer trap driving expression in the root cap (line Q0171) in a *phf1-1* background. The specific complementation of the root cap was obtained by introducing the UAS-PHF1 construct, producing the *phf1* Q0171>>PHF1 line. The proper targeting of Pi transporters to the PM in the root tip of resulting plants was confirmed by introducing the fluorescent marker mCherry fused to the *PHT1;4* gene driven by the constitutive 35S promoter. This produced a strong fluorescent signal in the columella and lateral root cap, where the GFP marker driven by GAL4/UAS was also observed (*Figure 2A*). The proper targeting of the PHT1;4-mCherry fusion protein was validated by its colocalization with the PM-specific FM4-64 dye (*Figure 2—figure supplement 1A,B*). This confirms that an effective restoration of PHT1 targeting in the root cap PM has taken place in the *phf1* Q0171>>PHF1 line (*Figure 2—figure supplement 1A,C*). A very low, diffuse fluorescence signal could also be detected in other tissues (*Figure 2—figure supplement 1C*), corresponding to a previously reported low level of ER-retained protein in the *phf1-1* mutant (*Gonzalez et al., 2005*; *Bayle et al., 2011*). The root cap specificity of the *phf1* Q0171>>PHF1 complementation therefore provides a unique opportunity to investigate the effect of localized Pi uptake in the root cap with physiological studies.

Visualization (*Figure 2B*) and quantification (*Figure 2C*) of Pi absorption at the root apex of the *phf1Q0171>>PHF1*, *phf1-1* and WT lines were performed by real-time imaging. Our time course experiment revealed rapid Pi absorption after only 1 min following $^{33}$P addition (*Figure 2B*). We observed full restoration of Pi absorption at the *phf1Q0171>>PHF1* root tip similar to the WT level,

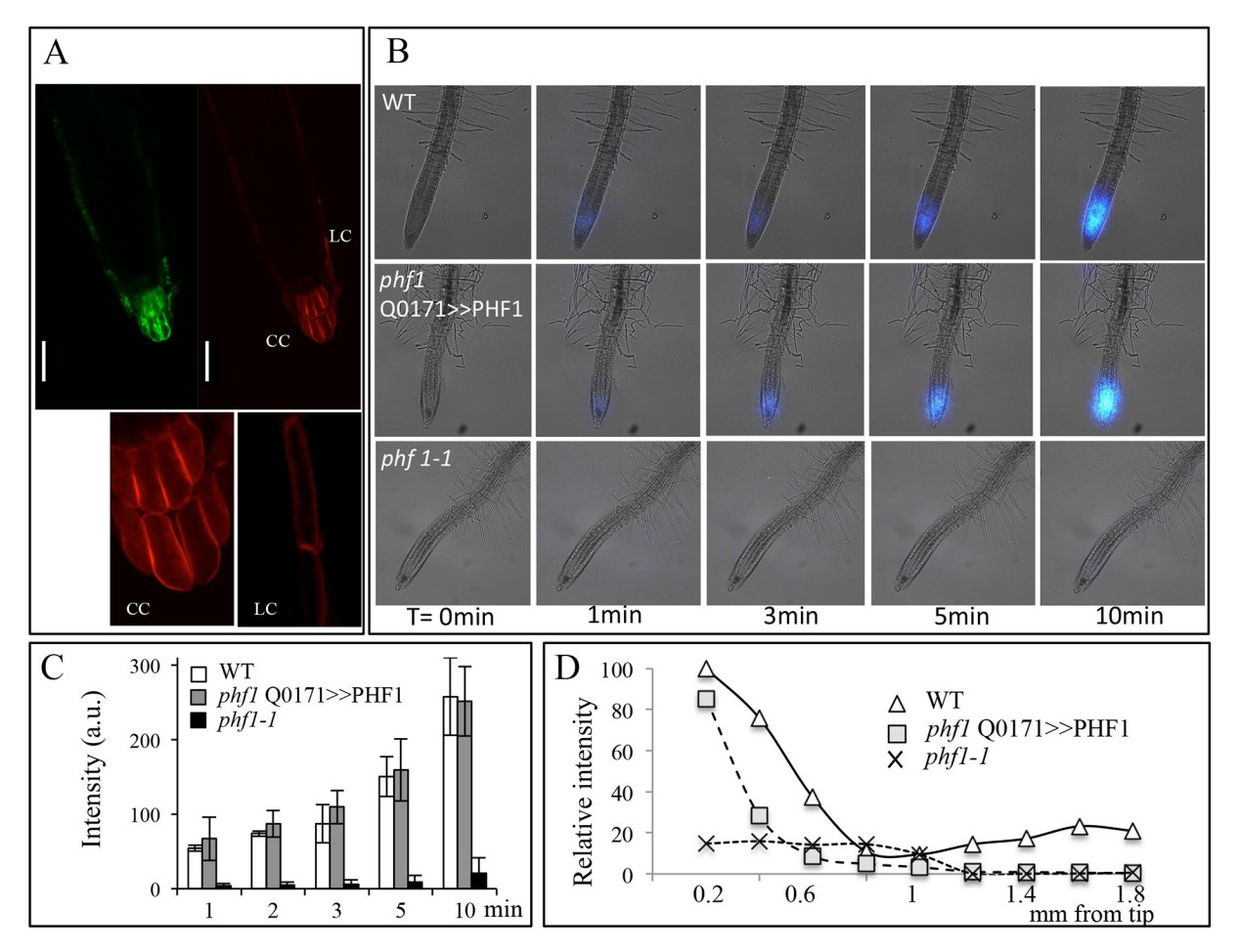

**Figure 2.** Root cap complementation of the *phf1-1* mutant in the *phf1* QO171>>PHF1 line. (**A**) Accumulation of PHT1;4 in the plasma membrane (red; 35S:PHT1;4-mCherry) correlates with PHF1 complementation in cells expressing GFP. Scale bars: 50 µm. Lower panels display magnified views of the mCherry image in the columella (CC) and lateral root cap (LC) cells. (**B**) Visualization of $^{33}$P absorption (blue) by real-time imaging. The image displays a time course during a 10 min period. (**C**) Quantification of radioactivity (200 kBq application) in the root apex (0.2 mm from the tip). Values are means ± SD. 3 root tips were analyzed. (**D**) Quantification of $^{33}$P along the root after 1 min. Measurements were taken from the tip to a distance of 1.8 mm at 0.2 mm intervals. A representative graph is shown. The experiment was performed eight times giving the same trend.

The following figure supplement is available for figure 2:

**Figure supplement 1.** Colocalization of the plasma membrane marker FM4-64 and PHT1;4-mCherry in the *phf1* Q0171>>PHF1 line.

whereas no labeling was detected in the *phf1-1* mutant during the same time period (*Figure 2B,C*), demonstrating the successful functional complementation of Pi uptake activity.

Real-time quantification of $^{33}$P uptake revealed a significant role for the root tip in Pi absorption as compared to the mature zone of the WT root (*Figure 2D*). WT plants exhibited a very low level of radiotracer (*Figure 2D*) in the region between these two parts of the root (corresponding to the elongation and differentiation zones; approximately 500–900 µm from the apex), indicating poor Pi absorption activity. Root labeling occurs in the differentiated zone (≥1.2 mm from the root tip) as a result of uptake through the epidermis. The signal present in the WT was impaired in both *phf1-1* and *phf1Q0171>>PHF1* (*Figure 2D*). This confirms the specificity of the root cap complementation in *phf1Q0171>>PHF1*, since the $^{33}$P quantification only describes a specific enrichment in the *phf1Q0171>>PHF1* root tip (*Figure 2D*). Similarly, in the WT, the accumulation of P33 was high in the root tip.

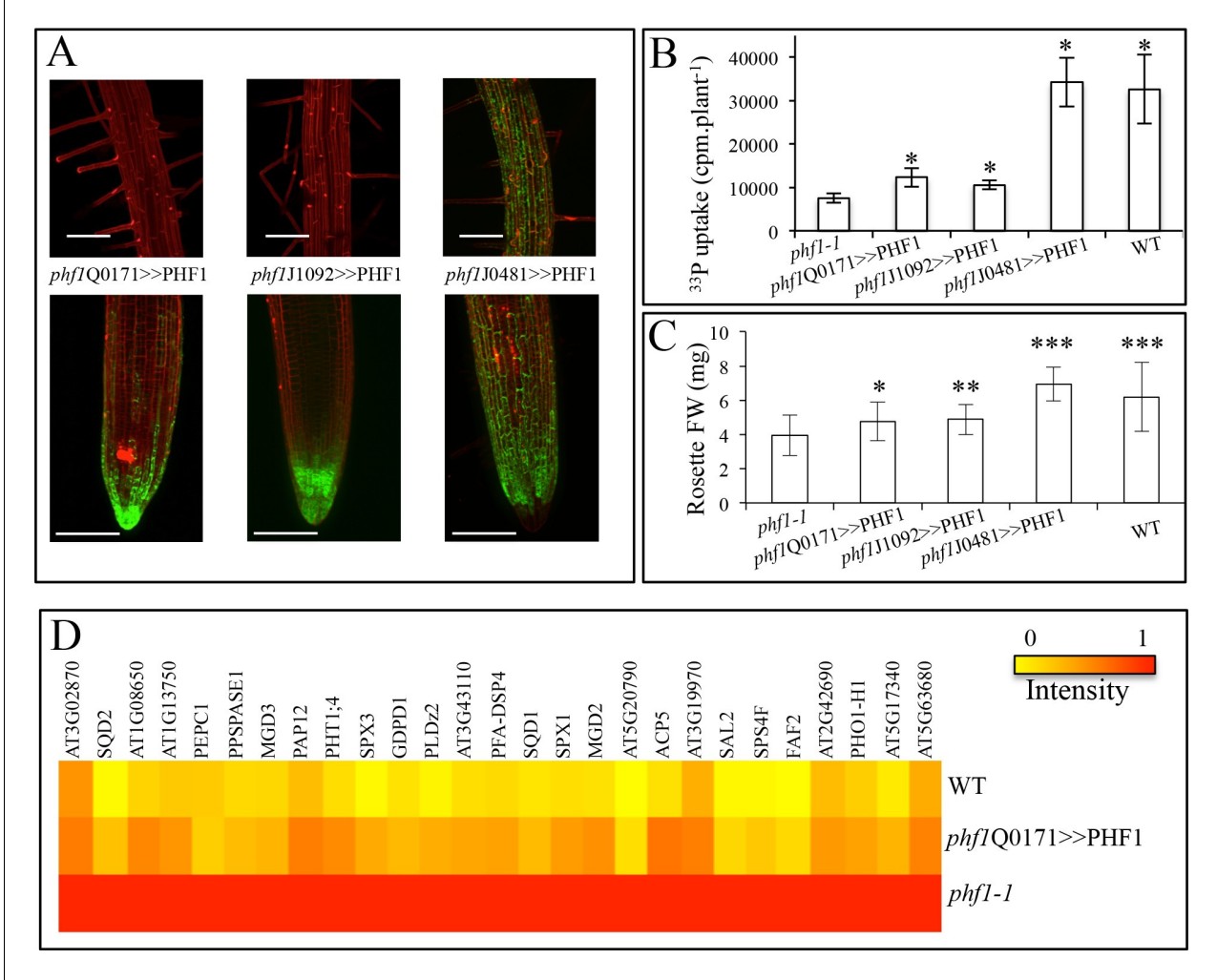

**Figure 3.** Effect of tissue-specific PHF1 complementation on plant physiology. (**A**) GFP expression pattern (green) labels root tissues (root tip and mature zone) where *PHF1* complementation occurs in three different transgenic lines. Scale bars: 100 μm. (**B**) Quantification of Pi uptake after transfer of plantlets to +P medium for 3 d in the presence of $^{33}$Pi. Values are means ± SD. 10 plantlets were analyzed individually. Significantly different from *phf1-1*: P<0.0001 (Student's t-test). (**C**) Rosette biomass. Plants were grown in -P and transferred to +P medium for 4 d. Values are means ± SD of 23 to 24 individually weighed rosettes. Significantly different from *phf1-1*: *P=0.008; **P=0.004; ***P<0.001 (Student's t-test). (**D**) Expression level of Pi starvation markers after a 3-day transfer to +P medium relative to low Pi. Results are normalized to *phf1-1* values according to *Figure 3—figure supplement 3*.

The following source data and figure supplements are available for figure 3:

**Source data 1.**

**Figure supplement 1.** Effect of PHF1 complementation on free Pi content.

**Figure supplement 1—source data 1.** Effect of PHF1 complementation on free Pi content.

**Figure supplement 2.** Effect of the PHF1 complementation on the expression of the Pi-starvation marker PHT1;4 as revealed by fusion of the GUS reporter gene to the PHT1;4 promoter.

**Figure supplement 3.** Effect of PHF1 complementation on gene expression.

**Figure supplement 3—source data 1.**

# Pi uptake *via* the root cap alters plant physiology

## Contribution to plant growth

Reduced growth in the aerial parts is a well-known consequence of Pi starvation. This feature is conserved in the *phf1-1* mutant (*Gonzalez et al., 2005*) even in Pi-rich medium, in agreement with the low Pi-status of these plants. By contrast, the root architectures of *phf1-1* (*Thibaud et al., 2010*) and the WT were similar in Pi-rich medium, confirming that root growth responds to external Pi (*Svistoonoff et al., 2007*; *Peret et al., 2011*). This facilitates analyses between the *phf1-1* mutant, the complemented line and the WT control at the whole plant level. Free Pi measured in roots or leaves of the complemented line was similar to the *phf1-1* mutant, indicating full metabolization of the excess absorbed Pi (*Figure 3—figure supplement 1*, *Figure 3—figure supplement 1—source data 1*).

In addition to the *phf1Q0171>>PHF1* line (in which complementation only takes place in the lateral root cap and columella), two other complemented lines were examined (*Figure 3A*). Complementation in the columella and the epidermal initial cells was possible with the *phf1J1092>>PHF1* line, whereas *phf1J0481>>PHF1* allowed PHF1 expression in all external root layers (*i.e.* the cap and epidermis). As expected, the *phf1J0481>>PHF1* line fully restored Pi uptake to a level similar to that of the WT. By contrast, Pi uptake in *phf1Q0171>>PHF1* and *phf1J1092>>PHF1* restored 20% of the difference in Pi uptake between the WT control and the *phf1-1* mutant (*Figure 3B*, *Figure 3B—source data 1*). It is likely that this value slightly underestimates the root cap's contribution to Pi uptake, as the *phf1-1* mutant probably displays residual activity in this tissue.

The effect of the increase in Pi absorption on biomass was also investigated. Rosette weight of in vitro plantlets was increased in all three complemented lines (as compared to the *phf1-1* mutant, *Figure 3C*, *Figure 3C—source data 1*) after transfer from Pi-depleted to Pi-rich medium during 4 days. The root cap complementation restored 40% of the growth (*phf1Q0171>>PHF1* and *phf1J1092>>PHF1* lines) and in agreement with the full restoration of Pi uptake similar to the WT control, the *phf1J0481>>PHF1* line does not present any growth defects.

To investigate the long term effect of the root cap complementation, we first grew plants on sand (hydroponically with 10 µM Pi supply) for 3 to 4 weeks; the experiment was terminated at the appearance of the first stem. The rosette biomass of the *phf1Q0171>>PHF1* line (in which complementation was restricted to the root tip) exhibited a significant 80% increase in rosette biomass, as compared to the *phf1-1* mutant (*Table 1*, *Table 1—source data 1*). Then, we investigated the growth of plants on soil exhibiting poor P content (less than 0.1% total P). A nutrient solution specifically depleted of Pi was used to water the samples every day. We observed a significant difference after 6 weeks of development, despite the high variability associated with this kind of experiment (as a result of heterogeneous soil composition). In this case, the gain in biomass of the root cap-complemented line turned out to be even higher, reaching 180% (*Table 1*) when compared to the *phf1-1* mutant. The hydroponics and soil experiments were replicated several times, confirming the significant contribution of Pi uptake through the root tip to plant development.

**Table 1.** Rosette biomass during long-term experiments. Plants were grown on soil or hydroponically in sand.

| FW (mg) | On soil | Hydroponically on sand |
|---|---|---|
| phf1-1 | 3.07 ± 2.23 | 3.04 ± 0.70 |
| *phf1* Q0171>>PHF1 | 8.66 ± 4.09* | 5.53 ± 1.56* |
| WT | 41.17 ± 24.76* | 11.04 ± 2.63* |

FW: fresh weight. Values are mean ± SD. Comparison to *phf1-1* line: *$P \leq 0.0009$ (Student's t-test). 10 to 23 rosettes were individually weighed.

**Source data 1.** Rosette biomass during long-term experiments.

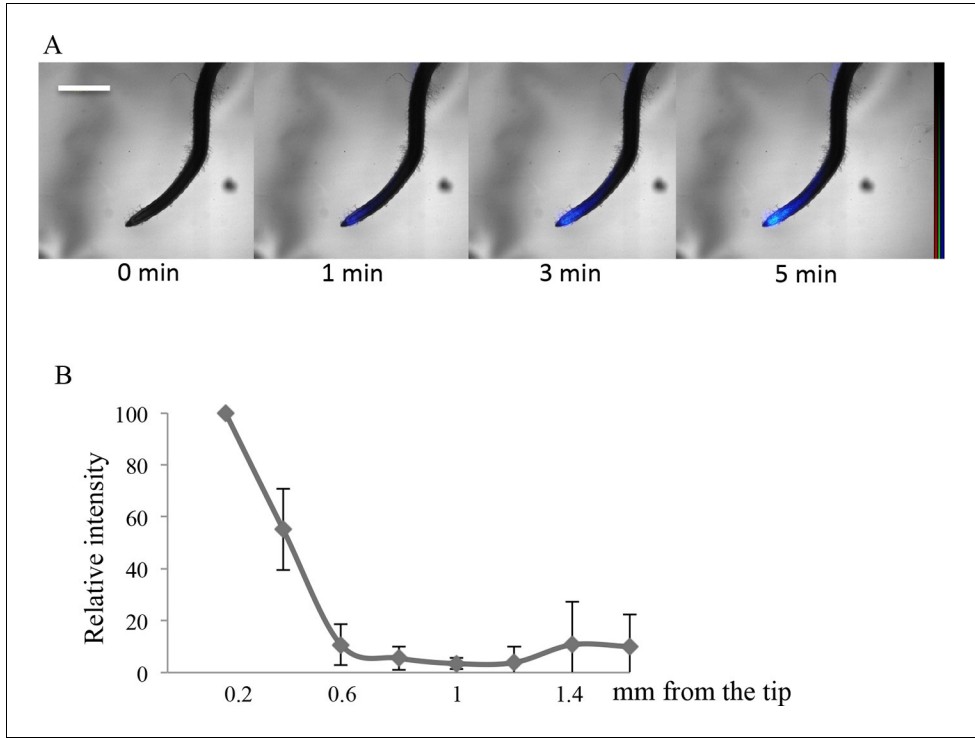

**Figure 4.** Imaging of $^{33}$P uptake at the root tip in *Oryza sativa*. (**A**) Time course of $^{33}$P uptake (200 kBq application). (**B**) Quantification of $^{33}$P (after 1 min absorption) along the root in 0.2 mm intervals, extending 2 mm from the tip. Values are means ± SD of 4 plantlets. Scale bar: 0.5 mm. $^{33}$P level is represented as color intensity.

The following figure supplement is available for figure 4:

**Figure supplement 1.** Imaging of $^{33}$P uptake at the *Lotus japonicus* root tip.

## Contribution to Pi sensing

Previous experiments have revealed that Pi starvation drives major transcriptional regulation in the plant genome (*Thibaud et al., 2010*). PHT1;4 is a well-established marker, and is highly induced under Pi deficiency (*Muchhal et al., 1996*; *Misson et al., 2004*). To determine its contribution to Pi sensing, we used a translational fusion construct comprising a GUS reporter gene associated with endogenous PHT1;4 (*Misson et al., 2004*). This marker was present in a previously isolated *phf1-2* mutant background, and was introduced into diverse enhancer lines (*Figure 3—figure supplement 2*). As a consequence, the complemented lines (*phf1J1092>>PHF1, phf1Q0171>>PHF1* and *phf1J0481>>PHF1*) lost the strong induction of PHT1;4-GUS expression observed in the *phf1-2* mutant when Pi is present in the growth medium. This complementation effect was confirmed at the molecular level by analyzing the expression of several Pi starvation markers previously identified in a whole-genome transcriptomic analysis (*Thibaud et al., 2010*). This approach differentiated systemic and locally responsive genes according to their respective response to internal Pi status or to available Pi present in the growth medium. Most of the markers regulated by external phosphate were not affected, and were observed to react identically in WT, *phf1-1* and *phf1Q0171>>PHF1* backgrounds (*Figure 3—figure supplement 3A*, *Figure 3—figure supplement 3A—source data 1*). Analysis of the systemically regulated Pi starvation-induced genes revealed a different situation, in which a general repression of these markers occurred after transfer of the WT and *phf1Q0171>>PHF1* lines into Pi-rich medium after 2 days (results not shown) or 3 days (*Figure 3D* and *Figure 3—figure supplement 3B*, *Figure 3—figure supplement 3B—source data 1*). As previously reported (*Gonzalez et al., 2005*), these systemic markers are strongly induced when the *phf1-1* mutant line is grown in high Pi (*Figure 3D* and *Figure 3—figure supplement 3B*), whereas these genes are repressed in the WT. This confirms the result obtained with the GUS reporter gene fused

with the systemically regulated PHT1;4 (*Figure 3—figure supplement 2*). The repression of the systemically regulated genes was partially rescued in the root cap PHF1-complemented line. This repression was equivalent to half the repression observed in the WT. This is not proportional to the additional Pi uptake related to root cap complementation (at least 20%), confirming the existence of a non-proportional relationship between Pi uptake, growth and systemic regulation.

## Occurrence of root cap Pi absorption in other plant species

In order to investigate whether a significant role for the root cap in Pi import also exists outside of *Arabidopsis*, we finally examined two very different plants: a monocot (rice) and the wild legume *Lotus japonicus* (Fabaceae). Clear labeling of the root cap was observed in these specimens by applying the aforementioned protocol consisting of a short pulse of $^{33}$P radiotracer (*Figure 4* and *Figure 4—figure supplement 1*). The migration of Pi is clearly visible in both plants beginning at the extremity of the root tip (1 min labeling) and moving gradually toward the differentiated tissues of the root (after 3 and 5 min in rice and lotus; *Figure 4A* and *Figure 4—figure supplement 1* respectively). The quantification of $^{33}$P after 1 min (*Figure 4B* and *Figure 4—figure supplement 1*) indicates that Pi uptake takes place in the root cap, as in *Arabidopsis*. These results provide evidence that Pi uptake at the root tip is a feature shared by several disparate plant species.

## Discussion

We have established the existence of a highly active Pi uptake process in the root cap through a combined approach utilizing genetics (*i.e.* conditional ablation and the specific complementation of Pi transport) with real-time imaging of radioisotopes. Our results indicate that the root cap contributes approximately 20% of the total amount of Pi absorbed from the medium by the roots. This Pi contributes to the plant's total Pi pool, significantly sustaining the overall development of the plant. This is demonstrated by the 80 to 180% increase in biomass production observed in root cap-complemented lines as compared to the *phf1-1* mutant, during the vegetative growth of plants in soil.

The *Arabidopsis* root cap therefore appears to be a "hot spot" for Pi absorption, despite its small surface area. Interestingly, this confirms the hypothesis of the 17[th] century plant anatomist Nehemiah Grew, who drew a connection between plant nutrition and the distal end of the root tip (*Grew, 1682*).

At the root tip, only the cap seems to play an important role in Pi acquisition. Indeed, the elongation zone (distal to the cap) displays poor Pi absorption despite its essential metabolic activity. Various factors can be proposed to explain this observation. For instance, the fast rate of root growth reduces the period spent by cells in this area by 6 to 7 hr (*Beemster and Baskin, 1998*), limiting the time allocated to protein production. Additionally, the cell elongation process in this region could contribute to diluting the activity of the produced PHT1 transporters. These features do not affect the root cap, which is located below the elongation and meristematic zones of the root and is composed of cells that are 10 to 20 times smaller than mature epidermis cells (*Beemster and Baskin, 1998*) and that have low turn-over (*Dolan et al., 1993*). This likely contributes to the high cellular concentration of PHT1 proteins in the membrane of this tissue.

The root cap structure of *Arabidopsis*, composed of 180–260 cells (*Dolan et al., 1993*), can appear simple in comparison to other plant species. For example, there are as many as 4,000–21,000 cells in the maize root cap (*Clowes, 1976*). This disparity can be explained by the presence of additional cell layers in the columella, with 20 in maize (*Iijima et al., 2008*) and 16 in rice plantlets (*Wang et al., 2014a*), in comparison to only 5 in *Arabidopsis* (*Dolan et al., 1993*). Although this promotes the formation of a slightly larger cap (see *Lotus japonicus* in *Figure 4—figure supplement 1*) with more cell layers, it does not significantly increase the external surface in contact with the medium. Despite these species-specific features, the root cap has also been observed to absorb Pi in rice and *Lotus*.

The high concentration of Pi transporters in the root cap could be due to the physicochemical properties of phosphate. Pi interacts strongly with many cations and exhibits a very weak diffusion coefficient in soil (from $10^{-12}$ to $10^{-15}$ m$^2$ s$^{-1}$; *Schachtman et al., 1998*). Pi mobility is consequently very low, which promotes a Pi-depleted zone around the root as a result of its absorption by plants (*Kanno et al., 2012*). This suggests that the root cap may provide a temporal advantage to the plant (« first come, first served »), as the Pi stock available to the growing root becomes limited. This

feature could therefore be beneficial in natural soil with low Pi availability and high Pi heterogeneity, which may explain the higher biomass gain observed in our soil experiments as compared to hydroponic conditions.

Other micro- or macronutrients may also be absorbed by the root cap, although their identification will require further analyses. Nevertheless, an absence of tools to monitor the presence and movements of specific ions could pose technical challenges. Real-time isotope imaging is one practical approach to circumvent this issue. However, the limited detectable range of elements such as conventional ß-ray and some gamma-ray or X-ray emitters (*Kanno et al., 2012*) prevents studying the dynamics of many important plant nutrients at the microscopic level.

Plant nutrient carriers are known to be highly redundant, which poses a bottleneck for genetic studies. For many nutrients, redundancy is not only linked to gene duplications, since different transporter families have been described for some nutrients (*e.g.* nitrate), reinforcing the difficulty in finding common regulators. Genetic tools that can impair the activity of a complete family of transporters remain very rare, and are currently restricted to Pi nutrition, such as the *phf1-1* mutant used here. Previous spatial gene expression studies have investigated the presence of ion transporters in the root cap for iron (IRT1; *Vert et al., 2002*), potassium (ATkT3/KUP4/TRH1; *Vicente-Agullo et al., 2004*), nitrate (NRT1-1; *Remans et al., 2006*) and boron (BOR1; *Takano et al., 2010*). For boron, accumulation of this element around the quiescent center was predicted using a model based on the distribution of several boron transporters (*Takano et al., 2010*) and confirmed by chemical analysis (*Shimotohno et al., 2015*), thereby indicating a role for the root tip in boron uptake.

Defects in the nitrate NRT1-1 (*Krouk et al., 2010*) or potassium TRH1 (*Vicente-Agullo et al., 2004*) carriers affect root morphology (gravitropism, modification of lateral root development, *etc.*). This is related to the ability of these proteins to competitively transport auxin. To date, their presence in the root cap appears to be linked to a developmental role in the adaptation of root morphology to the presence of nitrate or potassium in the environment. Nevertheless, the role of these proteins in nutrition remains to be elucidated.

In the case of Pi, it is well documented that the primary root growth is disconnected from Pi import and Pi internal status (for review see: *Svistoonoff et al., 2007*; *Thibaud et al., 2010*; *Peret et al., 2011*). The present study confirms this view, as Pi absorption by the root cap has not been found to modify root architecture, although it significantly affects plant Pi nutrition.

As we have shown, plant growth varies according to the medium used (in vitro, hydropony, soil) but in all cases restoring Pi absorption in the root cap promoted a significant gain in biomass. Indeed for plantlets grown in vitro, a 20% increase in Pi uptake generated a 40% gain in biomass for the root tip-complemented line within 4 days. For mature plants grown for a longer period, this gain increased and could even reach 180% as observed with soil experiments.

The level of Pi derived from root cap absorption (with regard to the size of this tissue) contributed also significantly to the modification of Pi homeostasis and to regulation of gene expression, as revealed by changes in the expression of several transcriptomic markers regulated by internal Pi status (*Thibaud et al., 2010*). This observation is in agreement with previous reports that internal Pi may be involved in Pi homeostasis regulation (*Liu et al., 1998*; *Lv et al., 2014*; *Wang et al., 2014b*; *Puga et al., 2014*). The transcriptional response was not linearly connected to Pi uptake, since absorption in the root cap of the complemented line restored half of the regulation of Pi starvation systemic markers. As previously shown, this confirms that a small amount of Pi absorbed by the root (at the root tip level, as shown in this study) can regulate gene expression in the whole plant (*Thibaud et al., 2010*).

Charles Darwin characterized the root tip as a plant brain, based on its role in environmental perception and gravitropism (*Darwin and Darwin, 1880*). Numerous studies since then have revealed how the root cap responds to various abiotic stresses including: light, modulation of root architecture in response to phosphate (*Svistoonoff et al., 2007*), and potassium or nitrate supply (*Arnaud et al., 2010*). The present study establishes essential roles for this tissue in Pi nutrition, as well as Pi homeostasis adaptation. Finally, this study demonstrates that the root cap contributes to the plant's adaptation to soil Pi presence in a disproportionate way, considering its diminutive size.

## Materials and methods

### Plant growth

Surface-sterilized seeds were sown in vitro on square Petri plates containing modified Murashige and Skoog medium diluted 10x (MS/10) (from *Arnaud et al., 2014*), supplemented with 2 µM iron and phosphate (0 or 500 µM NaH$_2$PO$_4$ for low-Pi (-P) orhigh-Pi (+P) respectively). Low-phosphate medium containing 13 µM Pi (Pi present in the agar) was supplemented with NaCl to maintain Na concentration at 500 µM. Plantlets were grown vertically in a growth chamber under a 16 hr photoperiod at 23°C:21°C (light:dark). Growth conditions are described in further detail for each experiment. Rice (*Oryza sativa* var. Nipponbare) and *Lotus japonicus* (var. MG-20) seeds were surface-sterilized then germinated in distilled water for 3 d. Seedlings were then grown hydroponically on -P medium for 10 days. Seedlings were grown in a growth chamber under a 16 hr photoperiod at 28°C for rice and 23°C for *Lotus japonicus*.

Nine-day-old in vitro Arabidopsis plantlets (in +P) were transferred to washed sand distributed in pots (12 to 24 plants per genotype). The nutrient solution (MS/10 supplemented with 10 µM Pi) was replaced every 3 to 4 days. The cambisol surface layer (0–30 cm) was air-dried, gently ground, and sieved (<2 mm) for experiments with soil. The composition of this loamy soil (36.2% silt, 16.5% clay, and 47.2% sand) was determined according to normalized methods (Soil Analyze laboratory ARRAS, NF EN ISO/CEI 17025: 2005, INRA, France) to contain (in mg. g$^{-1}$ DW): organic C: 25.4, N: 2.14, P: 0.89, Ca: 7.68, K: 9.75, Na: 4.96, Fe: 24, and Si: 348, with a pH (water) of 6.9. The soil was mixed with fine sand (1 part sand:2 parts soil, w/w) and distributed in pots. Nutrient solution (MS/10 without Pi) was provided every day by immersion (20 min). Seeds were sown on the soil surface, and 24 to 30 plantlets per genotype were kept after 10 days.

### Biomass production, Pi uptake and free Pi measurement

Plants were grown on Pi-depleted medium for 7 days and then transferred to +P medium for 4 days. All experiments were performed in triplicate. Ten rosettes were individually weighed for biomass determination.

Biomass production was also measured on mature plants while growing hydroponically in sand or in soil for 3 to 6 weeks. Rosettes were harvested when flower buds appeared and were individually weighed (10 to 23 rosettes).

Pi uptake in the whole plant was measured after transfer for 3 days to +P medium supplemented with $^{33}$P (5.5 kBq/mL). Ten plantlets were individually analyzed for radioactivity. A preliminary experiment demonstrated that the *phf1* UAS:PHF1 line and the *phf1-1* mutant displayed similar influxes (16.5 ± 4.7 and 14.2 ± 4.3%, respectively, as compared to the WT).

Free Pi in roots and leaves was measured after transferring plants from -P (7 days) to +P for 4 days. Pools of 10 to 20 plants were analyzed in triplicate. Frozen material was homogenized in a grinder (Mixer Mill MM400, Retsch; Germany), resuspended and homogenized in MES buffer (0.17 M, pH 5.8, 10 µL per mg of fresh weight). After centrifugation, the supernatants were analyzed in a 96-well plate and triplicates of 5 – 20 µL subsamples were diluted into a final volume of 145 µL. Phosphate content was then measured using a Malachite green protocol (*Misson et al., 2004*) that was modified such that 30 µL of each reagent were added, and measurements were performed at 595 nm with a microplate reader (Biorad, Model 3550; USA). Phosphate concentrations were calculated using a calibration curve (performed with a KH$_2$PO$_4$ solution) and were expressed per root or rosette fresh weight.

### DNA construct and transgenic material

An amplified product (see *Supplementary file 1A* for oligonucleotides) containing the *PHT1;4* promoter (2.6 kb upstream of the start codon) along with the UTR and genomic sequence was cloned into the pENTR/D-TOPO vector using a pENTR directional TOPO cloning kit (Invitrogen; USA). The cloned fragment was then transferred into the Gateway vector pGWB4 (*Nakagawa et al., 2007*) to create a translational fusion with s*GFP*. Homozygous lines (in a WS ecotype background) were selected and designated as PHT1;4-GFP lines.

The UAS:PHF1 construct was created by amplifying *PHF1* cDNA (see *Supplementary file 1A* for primer sequences). The resulting PCR product was cloned into the pGEMT vector (Invitrogen). *PHF1*

cDNA was cloned into the pBI-UAS-KNAT4 plasmid between the UAS and Nos terminator sequences (*Truernit et al., 2006*) using the *Bam*HI and *Sac*I restriction sites to replace the *KNAT4* gene. This construct was introduced into the *phf1-1* background (ecotype Col-0, *Gonzalez et al., 2005*) and homozygous transformants (referred to as *phf1* UAS:PHF1) were selected.

Enhancer trap lines (ecotype C24) were introgressed into the *phf1-1* background by at least 3 successive backcrosses. The selected GAL4 lines include Q0171 (specific to the lateral root cap and columella), J1092 (specific to the root cap and epidermal initial cells), and J0481 (specific to the root cap and epidermis). Each line was used to express mGFP5 under UAS promoter control. For each GAL4 driver, a homozygous line was crossed with the *phf1* UAS:PHF1 line. Successive self-crossings were performed to obtain homozygous *phf1* QO171>>*PHF1*, *phf1* J1092>>*PHF1*, and *phf1* J0481>>*PHF1* lines. These lines were then crossed with the *phf1-2 pht1;4–1* double mutant. This double mutant was previously obtained by EMS mutagenesis of the *pht1;4–1* mutant (*Misson et al., 2004*).

PHT1;4 cDNA was cloned without its stop codon (see Table S1 for primer sequences) into pENTR/D-TOPO, resulting in pEN L1-PHT1;4-L2. The mCherry sequence was amplified by PCR (see Table S1 for primer sequences) from pEN L3-mCherry-HA-L2 (Addgene; USA) and cloned into pDONR P2r-P3 using the BP reaction to produce pEN R2-mCherry-L3. A 35S:PHT1;4-mCherry fusion was produced by multi-site Gateway reaction (*Karimi et al., 2007*) using the LR reaction between the following vectors: pEN-L4-2-R1 (containing 35S; *Karimi et al., 2007*), pENTR/D-TOPO, pEN R2-mCherry-L3 and pB7m34GW. The product was then introduced into Col-0, *phf1-1* and the *phf1* QO171>>*PHF1* line. Successive self-crossings were performed to obtain homozygous lines.

*FCY1* (from *S. cerevisiae*) and *UPP* (from *E. coli*) were amplified by PCR (see Table S1 for primer sequences). After purification, an additional PCR was performed to obtain the *FCY-UPP* fragment, which was cloned into the pDONR207 vector using Gateway® technology, resulting in pEN L1-FCY-UPP-L2. A UAS:FCY-UPP fusion was created by multi-site Gateway reaction (*Karimi et al., 2007*) between the pEN L1-FCY-UPP-L2, pEN-L4-UAS-R1 and pB7m24GW vectors. The final construct was used to transform Col-0 plants. A homozygous line was subsequently selected and crossed to the Q0171 line (backcrossed 3 times in Col-0). Experiments were performed with plants from F1 seeds, referred to as Q0171>>FCY-UPP.

All plant transformants were generated by the floral dip method as previously described (*Clough and Bent, 1998*), following introduction of the construct into *Agrobacterium tumefaciens*.

## Whole-plant autoradiography

To visualize $^{33}$P absorption, 7-day-old Arabidopsis seedlings grown in high-phosphate medium were incubated for 1 day in liquid medium supplemented with $^{33}$P (400 Bq/mL) and exposed against an imaging plate (Fujifilm; Japan) for 4 days at -80°C. Radioluminographic images of the seedlings were then scanned using the FLA-5000 imaging analyzer (Fujifilm; Japan) and analyzed using Image Gauge v4.0 (Fujifilm; Japan).

## Imaging $^{33}$P allocation in roots

Plants were grown in +P medium for 7 days. The middle of the root was isolated from the medium with Vaseline and a 10-µl drop of $^{33}$Pi solution (1 µM Pi, including 200 kBq $^{33}$P) was applied for 30 min. $^{33}$P signal in the root was detected using the Micro Real-time Radio Imaging system (*Kanno et al., 2012*) with an EMCCD camera iXon3 888 (Andor; USA).

## Real-time imaging of phosphate uptake

Seven-day-old plantlets grown in -P medium were transferred to glass slides covered with 0.1% agar, containing 1 µM Pi solution supplemented with $^{33}$P (200 kBq/10 µL). Real-time imaging of $^{33}$P uptake was performed using the Imaging System as described above. Images were acquired after 1 to 10 min, and radioactive signal was quantified using the AQUACOSMOS software (Hamamatsu Photonics; Japan) on a selected root tip zone (0.2 mm).

A separate labeling procedure was used to quantify a broader part of the root (2 mm). Roots of 7-day-old plantlets were immersed in 1 µM Pi solution containing 2000 kBq $^{33}$P for 1 min. Samples were then rinsed in a solution containing 1 mM 2,4-dinitrophenol. This inhibitor of ATP production was used to block cellular metabolic activity and to prevent ionic movement (including $^{33}$Pi).

Successive images along the root were then recorded as described above. Radioactive signal quantification was performed with AQUACOSMOS on successive 0.2 mm portions starting from the root tip. 13-day-old rice and lotus plantlets were fed 0.5% agar, containing 1 µM Pi solution supplemented with $^{33}$P (200 kBq). Successive images were obtained during 5 min and radioactive signal was quantified after 1 min, as described above.

To analyze Pi uptake in plantlets treated with 5FC, seedlings were grown for 3 days on +P and transferred for one day to +P medium containing 3.87 mM 5FC or DMSO (as a control). Subsequently, samples were transferred to -P medium containing 5FC or DMSO for 3 days before imaging as described above.

## Effect of 5-fluorocytosine treatment on root development

Q0171>>FCY-UPP plantlets were grown in +P medium and then treated with 3.87 mM 5FC or DMSO for 4 days. Images of the plates were taken when transferred, and after 2 and 4 days. Primary root length was quantified using the ImageJ software with the NeuronJ plugin (version 1.46r, http://imagej.nih.gov/ij).

## RNA extraction and RT-qPCR

RNA extraction, purification, reverse transcription and qPCR analyses were performed as previously described (*Thibaud et al., 2010*). Primer efficiency factor was measured for each gene, and GapC1, ROC3 and AT1G32050 were used as reference genes. Primer sequences are provided in *Supplementary file 1B*.

## Reporter line staining and imaging

GUS staining was performed as previously described (*Misson et al., 2004*) on plants grown for 10 days in -P medium supplied with 2 µM FeCl$_2$. For visualization, seedlings were either placed in water and observed under a stereomicroscope (MZ16, Leica Microsystems; Germany) or between coverslips and photographed under the microscope (LMD6000, Leica Microsystems; Germany).

For luminescence imaging, seeds from the proPHT1;1:LUC line (*Castrillo et al., 2013*) were sown on the +P medium with 2 µM Fe. After 4 days, plants were transferred to fresh medium containing 5 µM Pi supplemented with 50 µM Luciferin (D-Luciferin Firefly Potassium Salt, Biosynth). Root tips were excised 7 days after transfer and placed between a cover slip and a thin film of medium containing 5 µM Pi with 50 µM Luciferin. Root tips were imaged using a dedicated luminescence microscope (Luminoview, Olympus) connected to a cooled back-illuminated CCD camera (IkonM, Andor).

For sGFP imaging in the PHT1;4:GFP and PHF1:GFP lines, plants were grown in -P medium for 11 or 4 days, respectively. Plantlets were incubated for 3 min in 20 µg/mL propidium iodide (PI) for cell wall staining (at room temperature).

For mCherry and mGFP5 imaging, plants were grown in -P medium (+10 µM Fe) for 12 days. Tips of secondary roots were observed by confocal microscopy. GFP, PI and mCherry imaging were performed on either a TCS SP2 (Leica; Germany) or LSM780 (Zeiss; Germany) confocal microscope. For confocal imaging, GFP was excited at 488 nm (argon laser line), and PI (or mCherry) was excited at 561 nm (diode-pump solid-state laser). Fluorescence was detected with the LSM780 confocal microscope using the following settings: sGFP (GaAsP, 491 – 545 nm), mGFP5 (PMT, 492 – 522 nm), mCherry (GaAsP, 607 – 696 nm), and PI (PMT, 586 – 685 nm). Detection of mCherry with the TCS SP2 confocal microscope was between 599 and 651 nm.

Characterization of mGFP5 localization in the enhancer trap lines (*phf1* QO171>>*PHF1*, *phf1* J1092>>*PHF1*, and *phf1* J0481>>PHF1) was performed on 5-day-old plantlets grown in -P stained with PI. Q0171>>FCY-UPP plantlets were grown for 7 days in +P and then treated with 3.87 mM 5-fluorocytosine (5FC) or DMSO (5FC solvent) for 5 days, or by heat shock at 95°C for 5 min in water. Plants were then treated with PI. Images were acquired with a macroscope equipped with structured illumination (Axiozoom V16, + ApoTome2, Zeiss; Germany). Fluorophores were excited using a mercury lamp (GFP: Ex filter 470 nm BP40, Em 525 nm BP50; PI: Ex filter 572 nm BP25, Em 629 nm BP62). Images are composed of maximum intensity z-series projections of 26 to 33 images at 2-µm intervals (software: Zen 2012, version 1.1.2.0, Zeiss).

Localization of the PHT1;4-mCherry fusion protein to the PM was analyzed in the *phf1* QO171>>*PHF1* background with 1 µg/mL FM4-64 for 5 min at room temperature (Interchim;

France). Co-imaging of FM4-64 and mCherry was performed on a LSM780 confocal microscope in spectral mode (Ex: 561 nm, Em range: 563 – 696 nm). Linear unmixing was applied to separate mCherry and FM4-64 components, using reference spectra acquired separately and the Zen software (Zen 2012 SP1, version 8.1, Zeiss). Fluorescence intensity profiles were subsequently calculated (software: Zen 2012, version 1.1.2.0, Zeiss).

## Statictical analysis

All statistical analyses were performed with GraphPad prism 6 software (version 6.0f). Values were tested for normality (D'Agostino-Pearson omnibus normality test) and variance homogeneity (Brown-Forsythe and Bartlett's tests). Unpaired t-test (Student's test) was run to compare treatments.

## Acknowledgements

We thank Drs. T Desnos and N Leonhardt for valuable suggestions and their kind revision of the manuscript, N Pochon-Brousse and S Deschamps for technical support and B Loveall of Improvence for language correction services. Lines were gratefully provided by Drs. L Laplaze (enhancer traps), V Rubio (PHF1:GFP), A Leyva (proPHT1;1:LUC) and J Paz-Ares (phf1-1 mutant). We thank C Sarrobert and C Latrille for providing the soil analysis and the GRAP team for plant culture assistance. This work was supported by funding from the French CEA (LN, MCT, SC, JFA, VB, HJ, ED, EM, SK), a Marie Curie International Reintegration grant from the 7th European Community Framework Programme (HJ), PIA-RSNR, DEMETERRES from the French government (MCT, LN), the Japan Society for the Promotion of Science (SK), the University of Tokyo (TN, SK), and the ANR REGLISSE project (LN, MCT). Microscopy facilities were funded by the European Union (ERDF), the Région Provence Alpes Côte d'Azur, the Conseil Général, the French Ministry of Research, and the CEA.

## Additional information

### Funding

| Funder | Grant reference number | Author |
| --- | --- | --- |
| CEA-TRANSRAD 2012 | Transrad 2012 | Satomi Kanno |
| JSPS-22880009 | 22880009 | Satomi Kanno |
| Marie Curie Reintegration grant-PIRG05GA2009 249173 | PIGR05GA2009 249173 | Hélène Javot |

The funders had no role in study design, data collection and interpretation, or the decision to submit the work for publication.

### Author contributions

SK, Conception and design, Acquisition of data, Analysis and interpretation of data, Contributed unpublished essential data or reagents; J-FA, Conception and design, Acquisition of data, Analysis and interpretation of data; SC, VB, BP, Acquisition of data, Contributed unpublished essential data or reagents; RB, Drafting or revising the article, Contributed unpublished essential data or reagents; HJ, Analysis and interpretation of data, Drafting or revising the article, Contributed unpublished essential data or reagents; ED, EM, Analysis and interpretation of data, Contributed unpublished essential data or reagents; TMN, Conception and design, Contributed unpublished essential data or reagents; M-CT, Conception and design, Acquisition of data, Analysis and interpretation of data, Drafting or revising the article, Contributed unpublished essential data or reagents; LN, Conception and design, Analysis and interpretation of data, Drafting or revising the article

## Additional files

### Supplementary files

• Supplementary file 1. (A) Primers used for constructs. (B) Primers used for RT-qPCR.

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
