## [Decision Letter]

[Editors’ note: a previous version of this study was rejected after peer review, but the authors submitted for reconsideration. The first decision letter after peer review is shown below.]

Thank you for choosing to send your work entitled "A novel role for the root cap in phosphate uptake and homeostasis" for consideration at *eLife*. Your full submission has been evaluated by Detlef Weigel as Senior editor and three peer reviewers, and the decision was reached after discussions between the reviewers.

We had a lengthy discussion among the reviewers and the Senior editor, and based on the discussion and the individual reviews below, we regret to inform you that your work in its current form cannot be considered for publication in *eLife*. However, there was substantial enthusiasm for the topic and the approach, and we would be very interested in a substantially revised manuscript that would demonstrate the importance of the root cap in older soil-grown plants. This is in line with *eLife* policy to invite outright revision only in cases where it appears likely that a revision can be submitted within two months or so. Because performing experiments with mature plants will likely take longer, we are declining the manuscript in its current form. Please feel free to reach out to Detlef Weigel if clarification of the decision is needed.

*Reviewer #1:*

It has long been noticed that promoters of nutrient transporter genes are active in the root tip and that nutrient transporter proteins localize to the membrane of columella cells, suggesting that the root cap might be involved in nutrient uptake. This great manuscript finally shows that the root cap is indeed involved in phosphate uptake, can account for 20% of the phosphate absorbed by the root and can contribute to biomass production. This is enormous given its small surface area in comparison to the root epidermis and root hairs. To demonstrate this, the authors elegantly combine radioactive phosphate pulse labelling with root cap specific enhancer trap-mediated ablation, mutant complementation and gene expression analysis in *Arabidopsis*. They also show that phosphate accumulates in the root tip of *Lotus japonicus* and rice, suggesting that phosphate uptake via the root cap is widespread in plants.

To improve the manuscript a few changes and additions should be made:

Figure 1: The Results text related to this figure does not clearly state how ^[32]^Pi was applied as compared to Figure 1—figure supplement 1, where it clearly says that it was supplied to the middle of the root system. Although it is described in the Materials and methods it would be helpful to add this to the main text to highlight the difference between the two experiments. Furthermore, the exact accumulation of ^[32]^Pi in the "distal part of the root tip including the meristematic zone" cannot be seen from Figure 1. Therefore Figure 1 should also be cited in support of this statement.

What happens if ^[32]^Pi is applied to the root tip only?

Figure legend 2A: for precision "expression of PHT1;4 in the plasma membrane" should be "accumulation of PHT1;4 and localization to the plasma membrane".

The GFP marker for GAL4/UAS activity in Q0171 is nicely visible in the columella in Figure 2 (similar to Truernit and Haselhoff, 2008) but not visible in the columella in Figure 1—figure supplement 2. The reason for this difference is unclear.

For many graphs the information given in the figure legend is incomplete and this varies from figure to figure. Missing information refers to how many biological replicates were used to generate the data, what the displayed value (mean, median…?) and the error bar (St. Dev., SE?) represent and whether a mean or the value of a single replicate is shown. This is the case for Figure 2, Figure 3, Figure 4, Figure 1—figure supplement 3, Figure 4—figure supplement 1. The authors should complete the missing information.

Discussion, Paragraph one: "demonstrating" should be "confirming".

Discussion, Paragraph one: It is discussed that the elongation zone does not contribute to Pi uptake. I do not see the data to support this. ^[32]^Pi does not appear to accumulate there but in the root tip. However, the root tip specific ^[32]^Pi accumulation is also observed if ^[32]^Pi is fed and taken up from the middle of the root system. Pi uptake via the elongation zone was not directly tested.

Reviewer #2:

This paper reports the role of the root cap on phosphate uptake. The authors use elegant conditional Ablation and Pi uptake imaging systems for their studies. Although an important role of the root cap in Pi uptake had been predicted by the strong expression of PTH1 transporters and other genes in this region, formal demonstration of a biochemical and physiological role of the root cap in Pi uptake was still lacking.

The paper present solid data supporting the conclusions presented in the paper, which are of general interest for plant biologists. There are a number of points that could be address to make this report more solid and with higher impact.

1) To really demonstrate that the root cap has an important role in Pi uptake and homeostasis, the authors should analyze the impact of root cap ablation in mature plants grown in soils amended with different levels of Pi or in soil with different levels of Pi availability. The data on Pi accumulation and biomass accumulation was done only with small plants grown under in vitro conditions, which significantly different from the conditions and physiology of mature plants grown in soil.

2) The authors assign an important function in Pi uptake to the root cap because mutants impaired in Pi uptake, except in the root cap, can still import a significant amount of Pi an sustain growth. However, the role of the root cap could be overestimated when it becomes the only site of Pi uptake. Therefore, it would have been better to complement *phf1-1* with PHF1 in all root except the root cap; in other words the exact opposite experiment than that reported in the paper. In this case if the root cap is indeed important, Pi uptake and plant growth should be reduced in lines which have Pi impaired only in the root cap. This would definitely demonstrate the importance of Pi uptake in the root cap. This could be done using a GAL4 enhancer trap driving expression in all epidermal layers of the root except on the root cap or by expressing and RNAi for PHF1 in the root cap using the Q0171 enhancer trap line.

Reviewer #3:

In the present manuscript Kanno et al. use a combinatorial approach (genetic conditional ablation and specific complementation of Pi transport together with radioisotope imaging) to show remarkable active Pi uptake in the primary root cap of *Arabidopsis* (as well as rice and *Lotus japonicus*). The important contribution (~20%) of the root cap to Pi uptake is not well acknowledged in the literature, although this uptake corresponds to the previously determined, high expression of PHT1 genes in the root cap, and confirms a hypothesis from the 17th century. The tools used in this study are elegant and innovative, although the biological message in the end is limited.

The fact that the Pi taken up via the root cap alters plant growth is as expected as the result that Pi taken up via the root cap reverts high expression of (systemically regulated) P-starvation induced (PSI) gene transcripts. This reviewer therefore fails to see clear, new insights into Pi sensing. This would be different with the demonstration that the root cap itself (as opposed to other root areas) is the predominant or only place where root P-status is sensed and changes in physiological or molecular phenotypes are initiated. This however cannot be clearly inferred from the experiments.

Besides this general assessment, this reviewer had no major criticisms of the experimental work or the conclusions drawn.

[Editors’ note: what now follows is the decision letter after the authors submitted for further consideration.]

Thank you for resubmitting your work entitled "A novel role for the root cap in phosphate uptake and homeostasis" for further consideration at *eLife*. Your revised article has been favorably evaluated by two reviewers, and the evaluation has been overseen by Detlef Weigel as the Reviewing and Senior editor. The manuscript has been improved but there are a few remaining issues that need to be addressed before acceptance, as outlined below:

Figure legend 2D: the authors should indicate at which time point after P33 application the measurements were taken.

The logical connection between the three sentences from paragraphs one and two, subheading “Functional elements of Pi uptake are localized within the root cap” is not very clear. Rephrasing would help.

Paragraph three, subheading “Quantification of Pi uptake by the root cap”: the meaning of the last fragment of the sentence "as in the WT" is not entirely clear. Do the authors mean to say: "Similarly, in the WT the accumulation of P33 was highest in the root tip"?

Subheading “Occurrence of root cap Pi absorption in other plant species”: The time for movement of P33 from the root tip towards the differentiated tissue is indicated for rice. For completeness it should also be indicated for *Lotus japonicus*.

The sentences in paragraph eight in the Discussion lack scientific rigor: "As we have shown, plant growth does not respond linearly to ion concentration in the medium. Indeed, a 20% increase in Pi uptake generated a 40 to 180% gain in biomass for the root tip-complemented line, depending on growth conditions and treatment duration". It starts with the ion concentration in the medium, but then argues with percent uptake and compares two experiments in which not all compared parameters have been determined. If I understand the manuscript correctly, the 20% increased Pi uptake in the root tip complemented line was determined along with the 40% gain in biomass. The percent Pi uptake in the sand and in the soil experiment was not measured but here only biomass was determined. As it is not clear whether uptake is the same in different growth substrates, I advise against discussing the biomass from the long-term experiments along with the uptake values from the short-term experiment. Furthermore, I would either discuss biomass increase relative to the ion concentration in the medium or to the uptake instead of opening the sentence with one parameter and then talking about the other.

---

## [Author Response]

[Editors’ note: the author responses to the first round of peer review follow.]

In accordance with your suggestions, we have performed in vivo experiments on adult plants in order to assess the validity of the phenotype reported in vitro. This was done by growing the plants under two different conditions: either on sand supplemented with low Pi-nutrient solution, or in soil exhibiting a very poor P content supplemented with a nutrient solution without Pi.

Although growing *Arabidopsis* plants on soil with a poor Pi content proved to be difficult, we are now able to present 3 independent experiments on soil and sand. These results are reproducible and confirm our in vitro experiments: Pi uptake through the root cap has a highly significant effect on the biomass, which increased 2 to 3 times in comparison to the control line.

Reviewer #1:

*Figure 1: The Results text related to this figure does not clearly state how ^[32]^Pi was applied as compared to Figure 1—figure supplement 1, where it clearly says that it was supplied to the middle of the root system. Although it is described in the Materials and methods it would be helpful to add this to the main text to highlight the difference between the two experiments.*

The text has been modified in the manuscript (‘pulses of radioactive tracer were applied by immersing the whole root system in ^[32]^Pi solution', Results, subsection “Functional elements of Pi uptake are localized within the root cap”, first paragraph) as well as the Figure 1 legend (‘Accumulation of ^[32]^P in the root tip of *Arabidopsis* plantlets. Whole roots were immersed in ^[32]^P- enriched solution for 1 day’).

*Furthermore, the exact accumulation of ^[32]^Pi in the "distal part of the root tip including the meristematic zone" cannot be seen from Figure 1. Therefore Figure 1 should also be cited in support of this statement.*

This has been modified according to the suggestion (we slightly changed the text to avoid introducing Figure 1 before 1B). Consequently, the original Figure 1 has become Figure 1 and vice versa.

*What happens if ^[32]^Pi is applied to the root tip only?*

We have tried to do this experiment with *Arabidopsis* and other plant species, but we couldn’t obtain any satisfying results. In fact, it was difficult to precisely apply Pi on the root cap.

Depositing a solution on such a small area (the *Arabidopsis* root cap is less than 0.2mm long) resulted in capillary effects, which prevented a precisely controlled application of Pi on the root cap. Moreover, the root cap is extremely fragile and was often damaged during the experiment.

*Figure legend 2A: for precision "expression of PHT1;4 in the plasmamembrane" should be "accumulation of PHT1;4 and localization to the plasmamembrane".*

This has been modified as suggested by the reviewer.

*The GFP marker for GAL4/UAS activity in Q0171 is nicely visible in the columella in Figure 2 (similar to Truernit and Haselhoff, 2008) but not visible in the columella in Figure 1—figure supplement 2. The reason for this difference is unclear.*

The images in Figure 2 were focused on the surface of lateral root cap cells. Accordingly, it is not possible to precisely view the columella cells, as they are not in the same plane.

*For many graphs the information given in the figure legend is incomplete and this varies from figure to figure. Missing information refers to how many biological replicates were used to generate the data, what the displayed value (mean, median…?) and the error bar (St. Dev., SE?) represent and whether a mean or the value of a single replicate is shown. This is the case for Figure 2, Figure 3, Figure 4, Figure 1—figure supplement 3, Figure 4—figure supplement 1. The authors should complete the missing information.*

This has now been performed as suggested by the reviewer.

*Discussion, Paragraph one: "demonstrating" should be "confirming".*

This has been modified as suggested by the reviewer.

*Discussion, Paragraph one: It is discussed that the elongation zone does not contribute to Pi uptake. I do not see the data to support this. ^[32]^Pi does not appear to accumulate there but in the root tip. However, the root tip specific ^[32]^Pi accumulation is also observed if ^[32]^Pi is fed and taken up from the middle of the root system. Pi uptake via the elongation zone was not directly tested.*

Figure 2 shows that during Pi uptake experiments, ^[32]^P accumulation is high in the root cap and lower in a zone that extends 1 – 2mm from the tip (corresponding to the root hair zone). In between, in the elongation zone, we see a clear reduction of signal, suggesting a very low activity associated with this area.

Reviewer #2:

*1) To really demonstrate that the root cap has an important role in Pi uptake and homeostasis, the authors should analyze the impact of root cap ablation in mature plants grown in soils amended with different levels of Pi or in soil with different levels of Pi availability. The data on Pi accumulation and biomass accumulation was done only with small plants grown under in vitro conditions, which significantly different from the conditions and physiology of mature plants grown in soil.*

We have performed experiments with mature (3 to 6 week-old) plants as suggested by the reviewer. Plants were grown in sand supplied with a low Pi-nutrient solution (10μM Pi) or in soil without any Pi supply (using a Pi-deprived nutrient solution). Plants were grown in low Pi in both experiments. This revealed strong differences between the *phf1-1* mutant, the complemented line and the WT, which were abolished in high Pi. Results from experiments are reported in Table 1 (experiments were performed in triplicate). Biomass production of the complemented line and the *phf1-1* mutant were clearly different in both soil and sand, with an increase in biomass ranging from 80 to 180% for the complemented line (as compared to *phf1-1*). These findings strengthen the results obtained with young in vitro plantlets (10 day-old plants).

*2) The authors assign an important function in Pi uptake to the root cap because mutants impaired in Pi uptake, except in the root cap, can still import a significant amount of Pi an sustain growth. However, the role of the root cap could be overestimated when it becomes the only site of Pi uptake.*

We are aware of this and that is why we quantified Pi uptake at the root tip by radio imaging (Figure 2) to demonstrate that ^[32]^P accumulation in the root tip is not increased in the complemented line (as compared to the WT).

*Therefore, it would have been better to complement phf1-1 with PHF1 in all root except the root cap; in other words the exact opposite experiment than that reported in the paper. In this case if the root cap is indeed important, Pi uptake and plant growth should be reduced in lines which have Pi impaired only in the root cap. This would definitely demonstrate the importance of Pi uptake in the root cap. This could be done using a GAL4 enhancer trap driving expression in all epidermal layers of the root except on the root capo r by expressing and RNAi for PHF1 in the root cap using the Q0171 enhancer trap line.*

We did not test complementation of the epidermal cells because most available promoters or enhancer trap lines do not allow study of the whole epidermis and/or often exhibit expression in the root cap. As for the other suggestion, using RNAi to eliminate PHF1 in the root cap with Q0171 is difficult to set up, since silencing via RNAi is often incomplete and heterogeneous between lines. In our experience, it has been easier and more accurate to observe a 20% increase in plants exhibiting low Pi content than a 20% decrease in plants with high-Pi context.

Reviewer #3:

*The fact that the Pi taken up via the root cap alters plant growth is as expected as the result that Pi taken up via the root cap reverts high expression of (systemically regulated) P-starvation induced (PSI) gene transcripts. This reviewer therefore fails to see clear, new insights into Pi sensing. This would be different with the demonstration that the root cap itself (as opposed to other root areas) is the predominant or only place where root P-status is sensed and changes in physiological or molecular phenotypes are initiated. This however cannot be clearly inferred from the experiments.*

We fully agree with the reviewer. Our main point of interest is the remarkable Pi uptake activity that occurs at the root cap level, relative to its size. We cannot conclude that Pi sensing only takes place in the root cap, but we are sure that Pi absorbed via the root cap is sufficient to trigger consistent physiological and molecular modifications, which confirms the key role of this tissue in Pi uptake and homeostasis. The root tip is neither the sole nor predominant place where Pi is absorbed and/or sensed, and we do not want to deliver such an erroneous message.

Besides this general assessment, this reviewer had no major criticisms of the experimental work or the conclusions drawn.

We have now tried to take into account all of the reviewer’s suggestions. We hope this will improve the paper’s clarity.

[Editors' note: the author responses to the re-review follow.]

*Figure legend 2D: the authors should indicate at which time point after P33 application the measurements were taken.*

Figure 2 legend has been modified: ‘(D) Quantification of ^[32]^P along the root after 1 min. Measurements were taken from the tip to a distance of 1.8 mm at 0.2 mm intervals. A representative graph is shown. The experiment was performed 8 times giving the same trend.’

*The logical connection between the three sentences from paragraphs one and two, subheading “Functional elements of Pi uptake are localized within the root cap” is not very clear. Rephrasing would help.*

We agree with the reviewer that the sentence concerning accumulation of PHF1 is not in logical connection with the sentences before and after. We rephrased the paragraph (paragraph one, subheading “Functional elements of Pi uptake are localized within the root cap”).

*Paragraph three, subheading “Quantification of Pi uptake by the root cap”: the meaning of the last fragment of the sentence "as in the WT" is not entirely clear. Do the authors mean to say: "Similarly, in the WT the accumulation of P33 was highest in the root tip"?*

We have modified the paragraph and added the sentence as suggested (paragraph three, subheading “Quantification of Pi uptake by the root cap”).

*Subheading “Occurrence of root cap Pi absorption in other plant species”: The time for movement of P33 from the root tip towards the differentiated tissue is indicated for rice. For completeness it should also be indicated for Lotus japonicus.*

This has been indicated as suggested: ‘after 3 and 5 min in rice and lotus; Figure 4 and Figure 4—figure supplement 1 respectively’ (subheading “Occurrence of root cap Pi absorption in other plant species”).

*The sentences in paragraph eight in the Discussion lack scientific rigor: "As we have shown, plant growth does not respond linearly to ion concentration in the medium. Indeed, a 20% increase in Pi uptake generated a 40 to 180% gain in biomass for the root tip-complemented line, depending on growth conditions and treatment duration". It starts with the ion concentration in the medium, but then argues with percent uptake and compares two experiments in which not all compared parameters have been determined. If I understand the manuscript correctly the 20% increased Pi uptake in the root tip complemented line were determined along with the 40% gain in biomass. The percent Pi uptake in the sand and in the soil experiment was not measured but here only biomass was determined. As it is not clear, whether uptake is the same in different growth substrates I advise against discussing the biomass from the long-term experiments along with the uptake values from the short term experiment. Furthermore, I would either discuss biomass increase relative to the ion concentration in the medium or to the uptake instead of opening the sentence with one parameter and then talking about the other.*

We agree with this comment and have introduced a few modifications in the text to clarify this. The sentence has been modified (paragraph seven, Discussion) in order to separate short-term and long-term experiments. In the short-term experiment, both Pi uptake and biomass production have been measured.